# The Impact of Defect Size on Bone Healing in Critical-Size Bone Defects Investigated on a Rat Femur Defect Model Comparing Two Treatment Methods

**DOI:** 10.3390/bioengineering11030287

**Published:** 2024-03-19

**Authors:** Andreas Kammerer, Frederik Alexander Hartmann, Christoph Nau, Maximilian Leiblein, Alexander Schaible, Jonas Neijhoft, Dirk Henrich, René Verboket, Maren Janko

**Affiliations:** Department of Trauma, Hand and Reconstructive Surgery, Goethe University Frankfurt, University Hospital, Theodor-Stern-Kai 7, 60596 Frankfurt, Germany; hartmann@med.uni-frankfurt.de (F.A.H.); c.nau@med.uni-frankfurt.de (C.N.); leiblein@med.uni-frankfurt.de (M.L.); alexanderschaiblex@gmail.com (A.S.); neijhoft@med.uni-frankfurt.de (J.N.); d.henrich@trauma.uni-frankfurt.de (D.H.); verboket@med.uni-frankfurt.de (R.V.); janko@med.uni-frankfurt.de (M.J.)

**Keywords:** critical-size defect, bone healing, induced membrane, Masquelet, Epiflex^®^, human acellular dermis (HAD), tissue engineering, rat defect model

## Abstract

Critical-size bone defects up to 25 cm can be treated successfully using the induced membrane technique established by Masquelet. To shorten this procedure, human acellular dermis (HAD) has had success in replacing this membrane in rat models. The aim of this study was to compare bone healing for smaller and larger defects using an induced membrane and HAD in a rat model. Using our established femoral defect model in rats, the animals were placed into four groups and defects of 5 mm or 10 mm size were set, either filling them with autologous spongiosa and surrounding the defect with HAD or waiting for the induced membrane to form around a cement spacer and filling this cavity in a second operation with a cancellous bone graft. Healing was assessed eight weeks after the operation using µ-CT, histological staining, and an assessment of the progress of bone formation using an established bone healing score. The α-smooth muscle actin used as a signal of blood vessel formation was stained and counted. The 5 mm defects showed significantly better bone union and a higher bone healing score than the 10 mm defects. HAD being used for the smaller defects resulted in a significantly higher bone healing score even than for the induced membrane and significantly higher blood vessel formation, corroborating the good results achieved by using HAD in previous studies. In comparison, same-sized groups showed significant differences in bone healing as well as blood vessel formation, suggesting that 5 mm defects are large enough to show different results in healing depending on treatment; therefore, 5 mm is a viable size for further studies on bone healing.

## 1. Introduction

Large bone defects pose an ongoing problem for patients, surgeons, and healthcare systems. While fractures of long bones can be successfully addressed through the use of methods of osteosynthesis such as nails, plates, screws, and external fixation, defects of a size above 2 cm tend to heal poorly and likely end up in non-union without addressing the defect itself [1,2]. Segmental defects of long bones often occur following severe or high-impact trauma, often accompanied by open fractures, severe soft-tissue damage, damage to blood vessels or nerves, and are prone to infection [3,4]. Besides the localised issues, patient-dependent factors like higher age, comorbidities, malnutrition, the consumption of alcohol, or smoking may increase the risk of poor bone healing [5,6]. The prevalence of non-union is between 2% and 30% for long bone fractures [7]. The treatment is often time-consuming and costly [8]. Bones of the leg are affected in a high number of cases, resulting in a loss of mobility. Karger et al. [4] presented a study of 84 cases of long-bone reconstruction surgery. The lower extremity was involved in 70% of the cases, and for 90% of the cases achieving union took a mean of 14.4 months and 6.1 surgeries in total. A longer healing time and multiple operative interventions means a longer loss of productivity for the patient, causing indirect higher costs [5]. All this must be taken into consideration before choosing the optimal treatment method.

Besides these factors varying heavily for different patients and bones, a general understanding of the biological requirements for successful bone healing is needed. These are described, for example, by Giannoudis et al. in their “Diamond Concept” [9]. Of great importance for fracture healing are osteoinductive stimuli, an osteoconductive matrix, osteogenic cells, mechanical stability, and blood supply. If one of these factors is lacking at the fracture site, this will highly likely result in a non-union [10].

The gold-standard of treatment for bone defects is augmentation via autologous bone grafts [9,11]. Like osteosynthesis, this brings osteogenic cells to the fracture side. Masquelet et al. [12] enhanced this method with a two-step approach, with great success. Using this method, defects of up to 25 cm can be successfully healed [13]. The first operational step involves radical debridement, addressing the surrounding soft tissue and placing a PMMA (Poly-(methyl methacrylate)) cement spacer in the defect. Surrounding the cement spacer, a membrane is formed, the so-called induced membrane or Masquelet’s membrane. After 8–12 weeks, a second operation is performed. An incision is made into the membrane, the PMMA spacer is removed, and the resulting cavity is filled with an autologous bone graft. The Masquelet’s membrane forms a bioactive chamber surrounding the defect, providing growth factors, promoting vascularisation, protecting the graft from being re-sorbed, and behaving as a boundary so fibrous or soft tissue has no access to the fracture side [12,14,15,16,17]. The membrane releases mediators like the VEGF (vascular endothelial growth factor), BMP-2 (bone morphogenetic protein-2) or the TGF-β (transforming growth factor β), which are known to promote bone healing and angiogenesis [16,17,18].

Although it produces good clinical results for fractures and non-union healing, the Masquelet technique takes an extra 8–12 weeks until the final operation. For patients, this might result in a longer loss of productivity and a deterioration in quality of life. For this reason, many research groups, including ours, are searching for ways to shorten this process. A human acellular dermis (HAD) graft was added to the defect side as a substitute for the Masquelet membrane, leading to similar bone healing in a rat model, if combined with spongiosa as a defect filling [19]. Furthermore, Leiblein et al. compared rats treated with the Masquelet two-step technique to rats treated with HAD, and tried to substitute the bone graft with scaffolds (β-TCP) as well as bone marrow mononuclear cells (BMC) [20]. Bones treated with HAD showed similar bone healing and biomechanical stability to bones treated with the Masquelet technique, suggesting that the addition of BMC could promote vascularisation.

Although all the studies conducted hitherto have produced promising results, the size of the segmental bone defect studied was either 5 mm or 10 mm, depending on the experimental setup of the studies. Doubling the defect size could impact the required healing time to a yet-unknown extent. 

It is still unknown if the healing capacity of this one-step treatment is comparable to the bone healing promoted by the use of the Masquelet technique in terms of the defect size.

In addition, rats treated using the Masquelet technique had angiogenetic stimuli from the fracture ends, the fracture haematoma, and from the surrounding induced membrane from the start, while rats treated with HAD only had access to stimuli released by the fracture ends and the fracture haematoma. This might lead to a reduced blood vessel count depending on the treatment method or the defect size. 

Therefore, it was the aim of this study to analyse whether the level of bone healing and angiogenesis of the bones remains comparable when a defect is treated with the Masquelet technique or human acellular dermis if the defect size is doubled. Paraffin-embedded femurs from our own previous animal studies were used for the analyses. In these studies, various aspects of Masquelet’s induced membrane technique and the suitability of HAD use were investigated using the established femoral defect model in rats.

## 2. Materials and Methods

### 2.1. Animals

The experiments were performed in accordance with the established and approved regulations set by our Institutional Animal Care and Oversight Committee (project no. FK/1075, FK/1057 and FK/1053; Regierungspräsidium Darmstadt, Germany). The experiments were conducted between 2015 and 2020 [19,20,21,22]. The samples gained from these studies were assessed and compared retrospectively. Rats were 8–10-week-old Sprague Dawley rats with a weight of approximately 250–320 g. All rats were male to exclude hormone-related bone healing differences in female rats. The rats were housed to a maximum of 4 rats per cage under the same conditions in the same facility.

The rats’ femurs were sorted into four groups. Groups Masq5 and HAD5 had femurs with a 5 mm defect size and were treated with either the induced membrane technique or HAD (Epiflex^®^, DIZG, Berlin, Germany) as a surrounding for the cancellous bone graft. Groups Masq10 and HAD10 were similarly sorted, with the only difference being a set defect size of 10 mm. The numbers of femurs examined in each group are shown in Table 1.

### 2.2. Human Acellular Dermis (HAD)—Epiflex^®^

For these experiments, Epiflex^®^ (The German Institute for Cell- and Tissue Replacement (DIZG gemeinnützige GmbH, Berlin, Germany), a human acellular dermis, was used for wrapping the critical-size defects examined in Groups HAD5 and HAD10. The acellular human dermis has been approved as a medicinal product (German Medicinal Products Act, §21). Epiflex^®^ is obtained from consenting, serologically screened donors and processed (via decellularisation and sterilisation) for use [23].

### 2.3. Surgery

The femurs of rats of a comparable age were used. The surgery was performed according to a standardised and well-established surgical procedure, with the femurs differing only in the defect size of 5 mm and 10 mm, as well as in the use of HAD (HAD5, HAD10) or the Masquelet technique (Masq5, Masq10). The surgical procedure is described in detail in the original studies that the femurs were taken from [24,25,26,27]. Male Sprague Dawley (SD) rats of between 8–10 weeks old with a body weight between 250–320 g were used (Janvier Labs, Saint Berthevin, France). The femurs were retrieved after 8 weeks of healing time. Only the bones showing no signs of infection or plate/screw loosening were included in further assessments. The obtained bones were stored at −80 °C. The analyses were performed according to the sequence µCT analysis (BMD) to examine biomechanical properties, histology, and immunohistology, as described in previous studies [19,20,21,22].

### 2.4. The Determination of Bone Union and Bone Mineral Density (BMD)

Bone union and BMD were assessed by means of µCT-analysis. These analyses were conducted in the previous studies [19,20,21,22], but were re-evaluated for the present study. The µ-CT-Scan was performed using a high-resolution in vivo micro-CT Skyscan 1176 (Bruker AXS, Karlsruhe, Germany). The long axis of the bones was aligned orthogonally with the X-ray beam (Al 0.5 mm; voltage: 50 kV; current: 500 µA; frame average: 7; rotation ra.: 180; rotation st.: 0.5), with the focus set on the defect. In doing so, two-dimensional pictures were captured, which could then be re-constructed and saved as 3D arrays. Bone mineral density was assessed using CTAn software (Bruker AXS), and bone union was assessed (as a percentage) as the bone volume of the total defect volume using InVesalius software (Version 3.1.1; CTI, Caminas-SP, Brazil).

### 2.5. Assessment of New Bone Formation

Movat pentachrome-stained histological slice specimens prepared as part of the previous studies [19,20,21,22] were used to assess new bone formation. The preparation procedure is described in detail in these studies.

High-resolution panoramic photographs were taken from the stained histological sections using a BZ-9000 microscope (Keyence, Neu-Isenburg, Germany) and BZ-Analyzer software (Keyence). The existence of fractions of newly formed bone, cartilage tissue, and fibrous tissue in the defect area was determined using ImageJ software “https://imagej.nih.gov/ij/ (accessed on 13 June 2021)” to describe bone healing. Based on the determined tissue fractions in the defect zones, a bone healing score analogous to Han et al.’s [24] was calculated. For this score, the proportion of bone or cartilage tissue is evaluated analogously to the size of the corresponding tissue fraction with a score value of 1 (below 10%) to 10 points (=90–100%), as depicted in Table 2. The proportion of fibrous tissue was assessed reciprocally. The lower the proportion, the higher the score value (maximum value = 10 points).

### 2.6. Assessment of Vascularisation

Blood vessels in the defect zones were detected using immunohistochemistry through the staining of the alpha-smooth muscle protein (α-SMA), which is expressed in the smooth muscle cells of blood vessels, using a mouse anti-rat-α-SMA antibody (AB_262054, clone 1A4, final concentration 2 µG/mL, Abcam, Cambridge, UK). The staining was performed as part of the previous studies [19,20,21,22], and the procedure is explained in detail therein. The histological preparations were again evaluated for the present analysis using ImageJ, and the area fractions of the blood vessels were calculated in relation to the defect area. The strategy for assessing blood vessel area fraction in the bone defect area using ImageJ is further illustrated in Appendix A.

### 2.7. Statistics

The statistics were calculated using SPSS (IBM, New York, NY, USA; Version 25). The differences between the groups were calculated using a single-factor analysis of variance (ANOVA) via the Tukey HSD post hoc test. The results were illustrated as boxplots with a median, as well as whiskers depicting the minimum/maximum values. *p*-Values of less than 0.05 were considered statistically significant (*).

## 3. Results

### 3.1. Bone Union and Bone Mineral Density

Comparing the bone union of the different groups, a significantly higher percentage of bone union can be observed in Groups Masq5 and HAD5 compared to Groups Masq10 and HAD10, which had a bigger defect size, as seen in Figure 1. Group HAD5 (median value = 93%, further depiction of median values in brackets) had a slightly higher median value than Group Masq5 (90%), although this was not statistically significant. Group Masq10 (60%) showed significantly better bone healing than group HAD10 (45%). An example of a µ-CT of a 5 mm bone defect treated with induced membrane technique is shown in Appendix A.

In contrast, a comparison of the bone mineral density depicted in Figure 2 showed no significant differences between the groups. The rats treated with the Masquelet technique (Groups Masq5 and Masq10) showed a non-significant trend towards higher bone mineral densities than their counterparts (Groups HAD5 and HAD10), which were treated with human acellular dermis (HAD).

No significant differences in bone mineral density were seen when comparing Masq5 (1.02 g/cm^3^) and Masq10 (1.08 g/cm^3^), and when comparing HAD5 (0.96 g/cm^3^) and HAD10 (1.02 g/cm^3^). Taking a closer look at the groups with the same sized defects, no significant differences could be found between Masq5 and HAD5, as well as between Masq10 and HAD 10. Noticeably, for both defect sizes, higher bone mineral density was obtained in the groups treated with the Masquelet technique compared to the groups treated with HAD for the same defect size.

### 3.2. Histology and Bone Healing Score

The Movat pentachrome-stained histological slices were compared using the bone healing score established by Han et al. [24]. An illustrative histological slice of group Masq5 is depicted in Figure 3. The rats with a 5 mm defect size which were treated with HAD (Group HAD5) showed the highest median bone healing score. Significantly higher values were observed in Group HAD5 compared to Group HAD10. A similar finding was seen when the Masquelet technique was applied, as the Masq5 group obtained a significantly higher bone healing score than the corresponding 10 mm group, Masq10.

Differences were also observed between the treatments for the 5 mm defect size. Thus, significantly higher scores were obtained in the groups treated with HAD, compared to defects of the same size which were treated with the Masquelet technique. A converse difference was seen in the groups with a 10 mm defect size, but the significance level was not reached (Figure 4).

When comparing the single parameters included in Han et al.’s bone healing score, differences between the groups can be seen as well. The Group HAD5 specimens formed significantly more bone tissue compared to all the other groups, with a mean of 66% bone tissue in the defect area (Masq5: 55%, HAD10: 20%).

Group Masq5 showed a significantly higher percentage of newly formed bone tissue (55%) compared to Group Masq10 (37%), whereas the percentage of cartilage formed was significantly higher for the 10 mm defects (Masq10: 14%, HAD10: 22%) compared to the corresponding 5 mm defects (Masq5: 6%, HAD5: 5%). No significant differences could be observed between groups with the same defect size.

The fibrous tissue fraction within the defect area was highest in Group HAD10 (14%), and was significantly increased compared to Groups HAD5 (3%) and Masq10 (7%). There was no significant difference between the groups with 5 mm segmental bone defects.

The area fraction of the parameter of the “remnant defect” was significantly lower in both 5 mm groups (Masq5: 35%; HAD5: 26%) compared to the corresponding 10 mm groups (Masq10: 43%; HAD10: 45%). Furthermore, the values of the HAD5 group were significantly lower than those of the Masq5 group (Figure 4).

### 3.3. Vascularisation

Vascularisation was determined histologically by means of a histomorphometric analysis of the α-SMA-stained slides.

The percentage of α-SMA-positive tissue was significantly increased in the HAD5 group compared with the other groups. Increased α-SMA area fractions were also measured in the defect area in the Masq5 group compared to the 10 mm groups, but the differences did not reach significance (Figure 5).

## 4. Discussion

The impacts of the defect size and the type of treatment on the healing and angiogenesis of the large femoral bone defects in the rat models were analysed in the present study. Bone specimens obtained in past studies were used in this retrospective analysis. Two related treatment modalities of large bone defects were compared. One was the Masquelet two-stage induced membrane technique, and the other was a single-stage membrane technique using a human acellular dermis (HAD) to separate the defect area from the surrounding tissue. Two defect sizes were compared: 5 mm and 10 mm. While the 5 mm defect more closely simulates the typical bone defect size of 3–6 cm for the application of the Masquelet technique [25], the 10 mm defect serves as a model for very large bone defects of more than 10 cm. The defect filling in each case was vital bone tissue obtained from syngeneic donor rats.

The comparisons revealed that the 5 mm defects had a significantly higher osseous coverage rate compared with the 10 mm defects, regardless of whether the Masquelet technique or the single-stage membrane technique was used. This finding was supported both radiologically and histologically. Comparisons between the 5 mm defect groups showed that the bone healing outcomes, as measured by a histologic bone healing score, were significantly improved using the one-stage HAD technique compared with the classic two-stage induced membrane technique.

In the 10 mm bone defects, on the other hand, significantly higher cartilage proportions and larger areas of fibrotic tissue were measured compared to the 5 mm bone defects. This finding indicates delayed bone healing in the 10 mm defects, which were probably still in the chondral phase of bone healing after the 8-week healing period. Angiogenesis in the defect area was most pronounced in the HAD5 group.

Masquelet has described cases of successful bone healing using the induced membrane technique for defects of up to 25 cm in size [11]. Though healing can occur even in these extreme cases, Piacentini et al. [15] have shown that in a case study of 18 long bone defects treated with the induced membrane technique, there was a significant difference in the mean defect length of 6 cm for successfully healed bones vs. a length of 10.3 cm for bones ending up in non-union.

Our µ-CT analysis showed close-to-complete union for both the groups with 5 mm defects, and approximately 60–45% union for both the groups with the larger defects, with a tendency towards better union for rats treated with the induced membrane technique. This might suggest poor bone healing for these bigger defects, but this is challenged by the results of a case study performed by Karger et al. [4] of 84 human long bone defects, where the mean union time was 14.4 months. After our set 8 weeks of healing time, it seemed like in bigger bone defects, bone union had not yet fully occurred, but bone formation and remodelling would continue to take place. Full bone union might be achieved later than in smaller defects. This assumption is supported by our observation that 5 mm defects showed a significantly higher percentage of bone tissue, while larger defects showed a significantly higher percentage of cartilage present at the defect side. Applied to the stages of bone healing, this might suggest that the smaller defects were already in the remodelling phase and achieved a certain degree of callus hardening, while the larger defects were still in the callus formation phase, producing fibrocartilage callus [26,27,28]. Chondrocytes produce this fibrocartilage callus, starting from the ends of the fractured bone, bridging the defect. This so-called soft callus acts as a scaffold for endochondral bone formation [26]. The larger gap of the 10 mm defects might take more time to bridge, so these defects might still have been stuck in this earlier bone healing phase, thus not providing the same degree of bone union. In conclusion, rat femur defects of 5 mm length seem to achieve sufficient bone healing after 8 weeks, whilst larger bone defects of 10 mm seem to need more time to heal and should thus be examined after 10–12 weeks of healing time. These findings might be applicable to humans, albeit more as a rule of thumb because of the heterogeneity of defects, which means that a large number of factors contribute to bone healing.

Interestingly, the groups treated with the with human acellular dermis as a replacement for the induced membrane technique histologically showed a trend towards better bone healing in comparison to the defects treated with Masquelet’s induced membrane technique, whereas the median bone mineral density of the defect site was slightly higher in the animals in the Masquelet groups. For the 5 mm defects, treatment with HAD led to a significantly higher bone healing score than treatment with the Masquelet technique. For the 10 mm defects, the bone healing score was generally lower, and the differences between the treatment strategies disappeared.

The same tendencies were observable in terms of the amount of newly formed bone tissue. The 5 mm defects treated with HAD showed significantly more bone formation than the defects treated with the Masquelet technique, while this was reversed for the 10 mm defects.

The results of the bone union examinations showed similar differences between the 5 mm and 10 mm defects. Both 5 mm groups showed significantly higher bone union than both of the 10 mm groups. This substantiates the hypothesis that larger defects need a longer healing time, although high-quality callus and bone are still produced, which might lead to complete bone formation if the healing time is extended.

The overall comparable bone healing of both treatment methods confirms the positive results of our previous studies using HAD as a substitute membrane [19,20]. However, the reason that HAD promotes better bone healing in smaller defects while forming less bone tissue in larger defects remains unclear. A possible mechanism might be the higher blood vessel density in the HAD5 group compared to all the other groups, as detected by α-SMA staining. The surplus of blood vessels, and therefore the better nutrition of the defect side, might be a key factor in explaining this better level of bone healing. For the 10 mm defects, the HAD group showed a non-significant increase in α-SMA-positive blood vessels, whereas the Masquelet treatment group showed significantly better bone formation at the defect side. To fully understand the processes that led to better healing in these defects, a further investigation of the local biological processes in the earlier stages of fracture healing is needed.

The treatment of infected or dirty wounds and fractures might remain an area best addressed by the induced membrane technique. The option to add antibiotics to the cement spacer and the biological properties of the induced membrane are lacking when using HAD. The use of HAD adds an artificial layer to the fracture side, sealing off the fracture, and might even encapsulate a possible infection. Though HAD had some promising results, the advantages of the induced membrane technique outweigh them in those cases.

What is essential for the further improvement of treatments of large bone defects is an understanding of biological processes. There is evidence that the induced membrane acts similar to the periosteum [29]. The induced membrane likely provides growth factors and mesenchymal stem cells and supports vascularisation [17,30]. The already promising results found when using HAD as a replacement for the induced membrane might be further improved by the addition of a cellular component. The addition of cell therapy to different treatment techniques of large bone defects in rats has already shown promising results. Seebach et al. [31], in loading a β-tricalcium-phosphate (β-TCP) scaffold with mesenchymal stem cells (MSC) and endothelial progenitor cells (EPC), showed an improvement in early vascularisation in a rat femur defect model. Nau et al. [32] used a β-TCP scaffold loaded with MSCs and EPCs and combined it with a periosteal flap to treat critical-size defects in rats. The combination of MSCs and EPCs resulted in the highest bone mineral density of any treatment option and created biomechanical stability. The disadvantages of these sorts of cells are their long and expensive culture conditions [33]. Bone marrow-derived mononuclear cells (BMC) can provide a faster and safe substitute for MSCs and EPCs, while having similar effects on large bone defect healing [34]. Janko et al. [35] and Verboket et al. [33] have already shown the positive impact of the addition of BMC to different kind of scaffolds.

## 5. Conclusions

In terms of critical-size defects, the results of this retrospective histological analysis suggest that a differentiation between smaller and larger defects should be made. Surgeons should take this into consideration before they suggest further therapy or revision surgery if critical-size bone defects have not achieved union after 8 weeks. Bone union might still be achieved later, but healing could remain at an earlier stage of fracture healing and still see complete formation and the bridging of the defect with fibrocartilaginous soft callus. The use of human acellular dermis (HAD) led to better bone healing than the use of the well-established induced membrane technique in smaller defects, although in larger defects, it seemed to be an inferior technique. HAD could promote a potent barrier for excluding soft tissue, as seen in histological results. This finding might be transferable to other clinical fields of use, such as isolating nerves from adhesions and surrounding tissue. A successful substitution or filling of the induced membrane seems possible, particularly in the treatment of smaller critical-size defects, while the induced membrane technique still seems to be superior in large defects. For the rat model, groups of the same size showed significant differences in bone healing and angiogenesis, indicating that even 5 mm defects may be a feasible model of critical defect size in further studies.

## Figures and Tables

**Figure 1 bioengineering-11-00287-f001:**
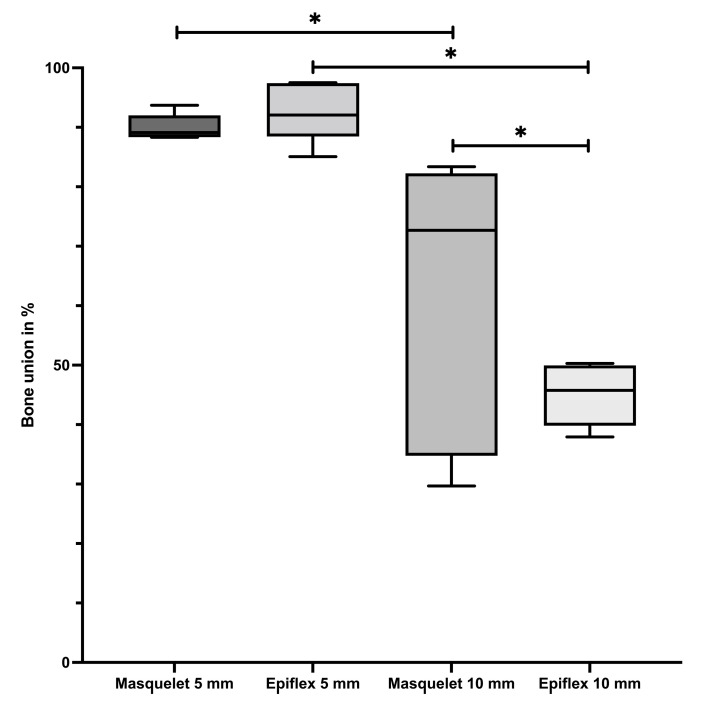
A significantly higher frequency of osseous unions in 5 mm bone defects. The graph illustrates the percentages of bone union achieved in Groups Masq5, HAD5, Masq10, and HAD10. * = *p* < 0.05.

**Figure 2 bioengineering-11-00287-f002:**
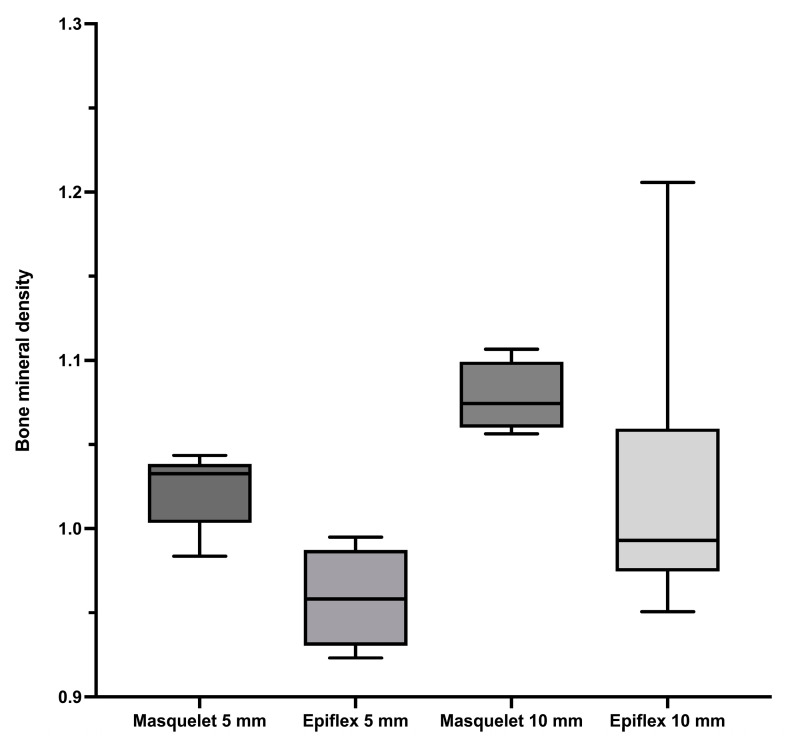
Higher bone mineral density was observed in the groups treated with the induced membrane technique than in the HAD-treated groups. The defect filling was a granular vital syngeneic bone material obtained from donor animals. There was a significant difference between the use of HAD5 and Masq10. The bone mineral density values for Groups Masq5, HAD5, Masq10, and HAD10 are shown.

**Figure 3 bioengineering-11-00287-f003:**
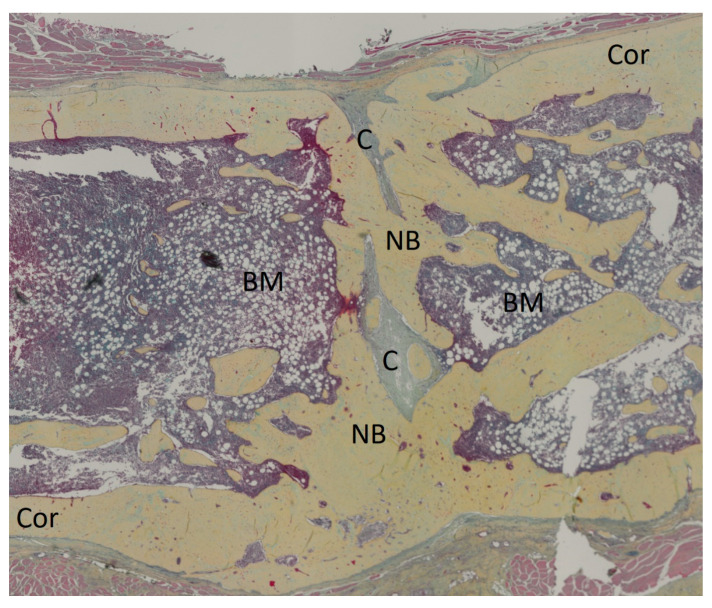
Illustrative histological slice of a sample of group Masq5 (5 mm defect treated using the induced membrane technique) stained with Movat’s pentachrome. Depicted are newly formed bone (NB), cartilage (C), bone marrow (BM), and cortical bone (Cor). Bony union was almost achieved.

**Figure 4 bioengineering-11-00287-f004:**
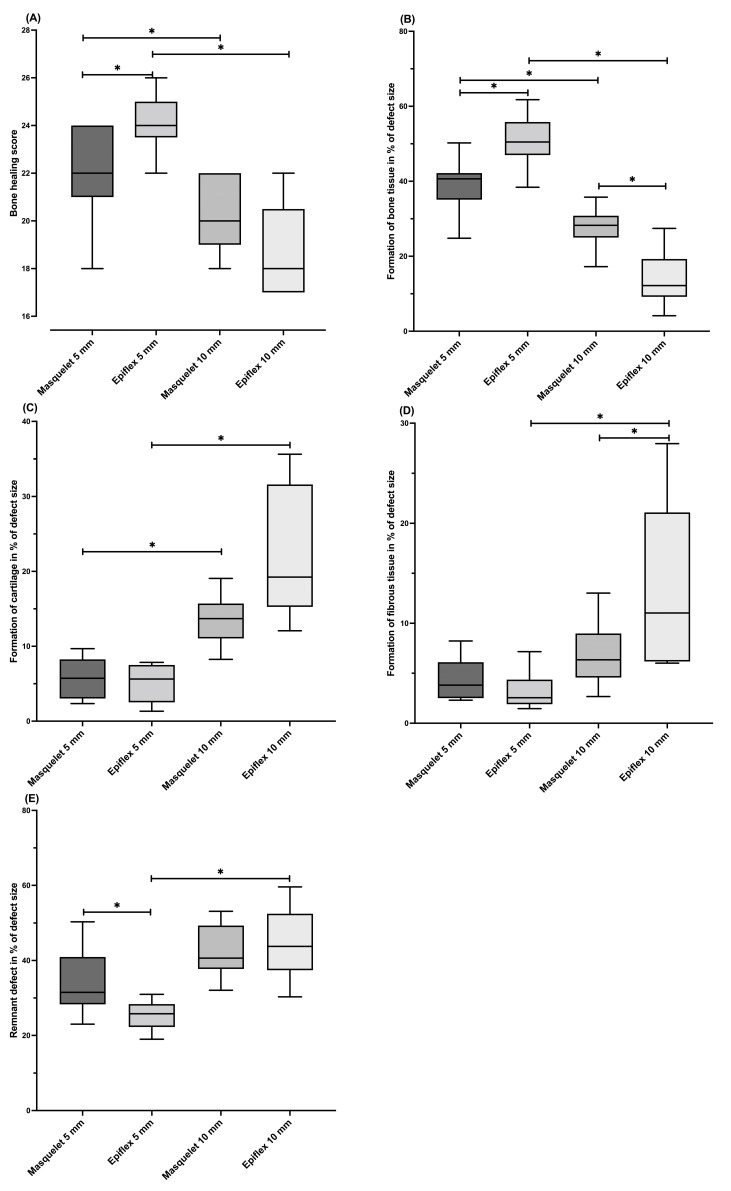
Histomorphometric evaluation of the Movat pentachrome-stained slices reveal significantly higher bone healing scores in Groups Masq5 and HAD5 compared to the corresponding groups, Masq10 and HAD10. In the groups with a 5 mm defect size, a significant surplus of new bone formation was observed. The groups with a 10 mm defect, in comparison, showed more cartilage formation, fibrous tissue formation, and a larger remnant defect size compared to the groups with 5 mm defects. (**A**) Bone healing score; (**B**) bone tissue formed; (**C**) cartilage formed; (**D**) fibrous tissue formed; and (**E**) remnant defect size. The bone healing score is given as an absolute number; (**B**–**E**) are given as percentages of the defect size for groups 1–4. * = *p* < 0.05.

**Figure 5 bioengineering-11-00287-f005:**
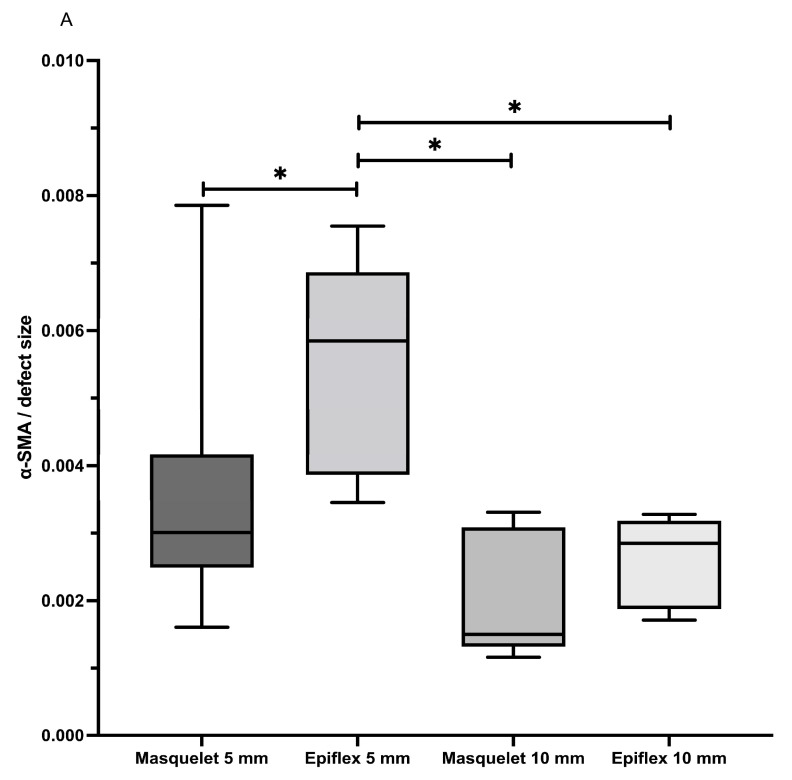
Evaluation of blood vessel density in the defect zone using a-sma staining (**A**). Values are given as area fractions of bone defects as evaluated by histomorphometric analysis of immunohistological stainings. * = *p* < 0.05. Representative immunohistological staining of α-sma positive blood vessels is shown in (**B**). Magnified images marked by the dotted line show enlarged details. Light blue dotted lines mark the borders of the defect. BM = Bone marrow, HAD = human acellular dermis, Ind.M. = induced membrane, and NB = new bone. Red arrows mark representative α-sma-stained blood vessels. Red bars represent a distance of 1 mm in 5 mm defects, of 2 mm in 10 mm defects, and of 300 µm in detail enlargements. Isotype control to confirm the specificity of α-sma staining is shown in (**C**). A 10 mm sized bone defect treated with the induced membrane technique is shown. No stained vessel structures are visible. mu = muscle tissue; red dotted lines indicate defect borders; and the red bar represents a distance of 2 mm.

**Table 1 bioengineering-11-00287-t001:** The number of rats included in each group (Groups Masq5, HAD5, Masq10, HAD10), the sample numbers in each group used for the assessment of histology, µ-CT assessing bone union (bone volume/total defect volume), bone mineral density (BMD), and α-smooth muscle actin (α-SMA).

Group Number:	Histology	µ-CT	α-SMA	BMD
(1). Masq5 (Masquelet + 5 mm defect)	11	5	12	5
(2). HAD5 (HAD + 5 mm defect)	10	5	11	4
(3). Masq10 (Masquelet + 10 mm defect)	13	8	7	4
(4). HAD10 (HAD + 10 mm defect)	6	7	7	6

**Table 2 bioengineering-11-00287-t002:** The bone healing score established by Han et al. [24]. The samples were given 4 to 10 points depending on the score they receive in each column. The number of points given for each column can be seen in the column on the left. The range is 0–40 points, with 40 being the highest possible score.

Score Value	Newly Built Bone	Newly Built Cartilage	Newly Built Fibrous Tissue	Remaining Defect Size
0	No visible formation	No visible formation	Fully filled with fibrous tissue	100%
1	<10%	<10%	<90%	<90%
2	<20%	<20%	<80%	<80%
3	<30%	<30%	<70%	<70%
4	<40%	<40%	<60%	<60%
5	<50%	<50%	<50%	<50%
6	<60%	<60%	<40%	<40%
7	<70%	<70%	<30%	<30%
8	<80%	<80%	<20%	<20%
9	<90%	<90%	<10%	<10%
10	Fully healed	Fully healed	No visible formation	No remaining defect

## Data Availability

Data are contained within the article and Appendix A.

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
