# Peer review of "The Impact of Defect Size on Bone Healing in Critical-Size Bone Defects Investigated on a Rat Femur Defect Model Comparing Two Treatment Methods"

_bioengineering, 2024, doi:10.3390/bioengineering11030287_

Round 1

Reviewer 1 Report

Comments and Suggestions for Authors

This manuscript compares the outcomes of 0.5 versus 1 cm non-union femoral defect in rats, re-analizing previously published data. Although there is a mild interest in publishing this data, which highlights the fact that the model behaves differently and may produce very different outcome depending on the lenght of these defects, I do not consider that it has enough novelty to justify its publication as a full article in Bioengineering, considering that apparently no original unpublished data is included in the analysis.

Also, the fact that all the data presented was already published in other articles is not stated clearly enough throughout the manuscript. When I was in a similar situation of re-publishing previously reported data but analyzed from a different angle as the starting point of new studies, I made sure to include this information in the figure legends and results section. In this article, this fact is briefly and partially referred to in the introduction (as "Paraffin-embedded femurs from our own previous animal studies were used for the analyses"), but this reference is very partial, as also the uCT analysis was previoiusly done, as well as the histology and immunohistology. So it seems that no novel analyses were carried out for this manuscript. It is only in the Materials and Methods section of this manuscript that this infromation is clearly provided. 

Besides this major issue, find below bulletpointed minor reviews:

P

Line 144: Provide more details about which ROI was used for the uCT analyses. The ideal would be to show a uCT sliceor 3D reconstruction showing the volume used as a supplementary file.

Line 180: Please provide an example supplementary figure of how the vascularization was assessed on ImageJ

Line 210:” The rats treated with the Masquelet technique (Groups Masq5 and Masq10) showed higher, but not statistically significant, bone mineral densities than their counterparts (Groups HAD5 and HAD10)”. Express it as a non-significant trend, you shouldn’t say “higher” because this is only true when stat significance is reached.

Figure 3: Please include one image per group. This figure is not referred to in the manuscript text.

Figure 4: Graph legends are too small.

Line 279: Representative images of each group alphaSMA immunohistochemistry and isotype negative controls should be shown in the manuscript.

Comments on the Quality of English Language

lease review grammar and language expressions throughout the manuscript. As an example:

Line 52: “Of great importance for fracture healing are osteoinductive stimuli, an osteoconductive matrix, osteogenic cells, mechanical stability, and blood supply via blood vessels. If one of these factors is lacking at the fracture side, this will highly likely result in a non-union [10]”. Here “via blood vessels” is obvious and hence unnecessary, and “fracture side” seems to be meant “fracture site”.

Reviewer 2 Report

Comments and Suggestions for Authors

The submission paper investigates the effect of defect size on bone healing in critical-size bone defects using a rat femur model, in comparison with the induced membrane technique with human acellular dermis (HAD). Major findings include that smaller defects (5mm) healed significantly better than larger defects (10mm) regardless of the treatment method. HAD used for smaller defects resulted in a significantly higher bone healing score and blood vessel formation compared to the induced membrane technique, suggesting its effectiveness in promoting bone regeneration in smaller defects.

The following comments could be considered to improve the manuscript:

1) The paper title is too long, it is suggested revising a shorter title to feature the major contributions.

2) In Section 2, some subsection titles should be added for better organization, such as 2.1 Animals, 2.2 Human acellular Dermis (HAD), 2.3 Surgery, etc.

3) With respect to figures 1, 2, 4 and 5, it is necessary to list the min-max, median values of the graphs in different tables, respectively.

4) Expand the discussion section to more thoroughly address the limitations of the study, including potential biases, the selection of animal models, and the generalizability of the findings to human clinical scenarios.

Round 2

Reviewer 1 Report

Comments and Suggestions for Authors

Dear authors,

Thank you for attending my revisions. My apologies for misunderstanding the lack of originality of the assays carried out on these samples. Since these are all new determinations I retract my previous recommendation to reject the manuscript and I recommend it for publication in its present form.

Reviewer 2 Report

Comments and Suggestions for Authors

The revised manuscript has been improved, and satisfied with the responses to my review comments.